# Revealing the mechanisms of membrane protein export by virulence-associated bacterial secretion systems

Lea Krampen[1], Silke Malmsheimer[1], Iwan Grin [1], Thomas Trunk[1,5], Anja Lührmann[2], Jan-Willem de Gier [3] & Samuel Wagner [1,4]

Many bacteria export effector proteins fulfilling their function in membranes of a eukaryotic host. These effector membrane proteins appear to contain signals for two incompatible bacterial secretion pathways in the same protein: a specific export signal, as well as transmembrane segments that one would expect to mediate targeting to the bacterial inner membrane. Here, we show that the transmembrane segments of effector proteins of type III and type IV secretion systems indeed integrate in the membrane as required in the eukaryotic host, but that their hydrophobicity in most instances is just below the threshold required for mediating targeting to the bacterial inner membrane. Furthermore, we show that binding of type III secretion chaperones to both the effector's chaperone-binding domain and adjacent hydrophobic transmembrane segments also prevents erroneous targeting. These results highlight the evolution of a fine discrimination between targeting pathways that is critical for the virulence of many bacterial pathogens.

[1] University of Tübingen, Interfaculty Institute of Microbiology and Infection Medicine (IMIT), Elfriede-Aulhorn-Str. 6, 72076 Tübingen, Germany. [2] Institute of Microbiology, University Hospital Erlangen, Wasserturmstr. 3-5, 91054 Erlangen, Germany. [3] Center for Biomembrane Research, Stockholm University, Svante-Arrhenius väg 16, SE-106 91 Stockholm, Sweden. [4] German Center for Infection Research (DZIF), Partner-site Tübingen, Elfriede-Aulhorn-Str. 6, 72076 Tübingen, Germany. [5] Present address: Section for Genetics and Evolutionary Biology, University of Oslo, Blindernveien 31, 0371 Oslo, Norway. Correspondence and requests for materials should be addressed to S.W. (email: samuel.wagner@med.uni-tuebingen.de)

Many pathogenic or symbiotic Gram-negative bacteria use type III secretion systems (T3SS) to inject effector proteins into eukaryotic host cells in order to promote bacterial survival and colonization[1]. *Salmonella enterica* serovar Typhimurium (*S.* Typhimurium) encodes two T3SS within its pathogenicity islands 1 and 2 (SPI-1/2)[2] that are essential for this pathogen's ability to invade and replicate within mammalian host cells[3]. Evolutionary and structurally related to bacterial flagella[4], T3SS are among the most complex protein secretion systems known. The central entity of the systems is a cell envelope-spanning macromolecular machine termed injectisome (Fig. 1a). It consists of a base that anchors the complex in the bacterial inner and outer membranes[5], of cytoplasmic components involved in targeting and preparation of substrates[6,7], of an inner membrane-embedded export apparatus facilitating substrate translocation located at the center of the base[8,9], and of a filamentous needle that protrudes from the bacterial surface and serves as conduit for substrates[10]. The entire system is composed of up to 20 different proteins with one to several hundred copies each[11].

Most substrates of T3SS are soluble proteins but each system exports at least one transmembrane protein that forms a pore for substrate translocation (translocon) in the host membrane (Fig. 1a). In addition, several other transmembrane effector proteins have been identified[12], amongst others the *Salmonella* SPI-2-encoded proteins SseF and SseG[13,14], the EPEC translocated intimin receptor (Tir)[15] and many inclusion membrane proteins (Incs) of *Chlamydia*[16]. Transmembrane domains (TMDs) within T3SS substrates pose a targeting conflict as two sequential secretion signals for two different incompatible pathways are concatenated in the same protein: an N-terminal T3SS signal and often also a chaperone-binding domain (CBD) guide export through the injectisome, however, transmembrane segments (TMS) signal inner membrane targeting[17]. The T3SS secretion signal comprises the first 20–25 residues of T3SS substrates. Its sequence is highly variable and enriched in polar but depleted in charged and hydrophobic amino acids, which is reflected in a lack of secondary structure. The role of the secretion signal in the targeting mechanism is still unclear[10]. The 20–50 aa long CBD is located downstream of the signal sequence. Cognate T3SS chaperones that are often encoded adjacent to their T3SS substrate bind to the CBD and guide the substrate to the sorting platform of the injectisome[7]. The chaperone is not secreted but stripped from the substrate by the action of the T3SS's ATPase[6]. Type III secretion is believed to be a mostly post-translational process because of the need for quick deployment of T3SS-effector proteins upon host cell contact, which requires bacteria to be charged with effector proteins. However, little is known about the relationship of translation with chaperone binding and substrate targeting. Instead, targeting and membrane integration of transmembrane proteins are mostly co-translational processes in bacteria[18]. A TMS emerging from the ribosome recruits the signal recognition particle (SRP)[17], which guides the ribosome-nascent chain complex to the SRP-receptor and finally to the Sec-translocon at the membrane, where translation proceeds and coincides with membrane integration of the TMS. In bacteria utilizing T3SS, post-translational T3SS-dependent secretion of substrates needs to be ensured even if these proteins expose a potential signal for co-translational membrane targeting and integration. This challenge of targeting discrimination is not only limited to T3SS but even more apparent for membrane proteins secreted through type IVB secretion systems (T4BSS) of Gram-negative bacteria, which contain mostly C-terminal secretion signals that can only act post-translationally. Targeting discrimination between membrane protein secretion out of the bacterial cell and insertion into the bacterial inner membrane is required to be effective as mis-targeting does not only lead to a loss of effectors but might also be suicidal for the

bacterium due to the pore-forming properties of some of the transmembrane effector proteins.

Here we investigate how discrimination between these membrane protein targeting pathways is achieved inside bacteria. We show that a balanced hydrophobicity of the TMS of T3SS substrates is one factor supporting targeting discrimination: While being sufficiently hydrophobic for principle membrane integration, targeting to the bacterial inner membrane is in general not facilitated by these segments. This observation was not only made for TMD-containing substrates of T3SS but also for many of T4BSS of *Legionella pneumophila* and *Coxiella burnetii* and thus seems to be a more general mechanism of targeting discrimination. A second factor for targeting discrimination, at least for some T3SS substrates, is the binding of their CBD and TMS by their cognate chaperones. This chaperone binding even prevents inner membrane targeting and insertion of TMS that are sufficiently hydrophobic for SRP-dependent targeting. Our results indicate that a TMS-specific co-translational targeting mechanism by T3SS chaperones prevents co-translational mistargeting such as by the SRP for subsequent post-translational secretion of membrane proteins through the T3SS injectisome.

## Results

**Hydrophobic T3SS substrates contain few TMS.** To gain a global view of the extent and quality of the secretion of transmembrane proteins by T3SS, we surveyed all previously identified T3SS substrates of *Aeromonas*, *Burkholderia*, *Chlamydia*, *Escherichia*, *Pseudomonas*, *Salmonella*, *Shigella*, *Xanthomonas*, and *Yersinia*[19] for their number of TMS by the prediction software ΔGpred using a sequence window of 18–35[20]. Of 174 scanned proteins in total, 36 were predicted to contain TMDs, of which 17 belong to the family of translocon components (Fig. 1b, Supplementary Data 1). For most of these proteins, only one or two TMS were predicted (17 and 17, respectively). Only two substrates were predicted to contain three TMS. Remarkable is the dominance of Inc proteins of Chlamydia among the T3SS TMD-substrates, of which we have only included the proteins of one sequenced isolate but of which a set of much greater diversity is known[16]. Over all, this analysis illustrates that T3SS commonly secrete TMD-substrates with effector functions other than translocators and suggests that T3SS favor the secretion of transmembrane proteins with TMDs of lower complexity.

**T3SS TMD-substrates are mostly of moderate hydrophobicity.** Helical-bundle membrane proteins have evolved such that the amino acid sequence of their TMS matches the physico-chemical properties along the cross-section of the lipid bilayer[20]. This match is reflected in the apparent ΔG ($\Delta G_{app}$) that is released upon integration of a TMS into the lipid bilayer. Besides the amino acid sequence, TMS length and amphiphilicity also contribute to $\Delta G_{app}$[20]. Based on experimentally determined values, $\Delta G_{app}$ can be calculated for each given polypeptide with windows of sequence lengths from 10 to 40 amino acids[20]. We analyzed the amino acid sequence of all 36 identified type III-secreted transmembrane proteins in order to determine whether the $\Delta G_{app}$ of their TMS would differ from those of classical single-span transmembrane proteins. In analogy to the work by Hessa et al.[20], we plotted the lowest $\Delta G_{app}$ for sequence windows of 18–35 (with length and amphiphilicity correction) in a histogram and compared it to previously determined values for single-span transmembrane and soluble proteins. The frequency of the $\Delta G_{app}$ of TMS of type III-secreted transmembrane proteins peaked at −1 kcal/mol, right in between the $\Delta G_{app}$ of single-span transmembrane proteins (−2 to −5 kcal/mol) and soluble proteins (2 to 4 kcal/mol) (Fig. 2a).

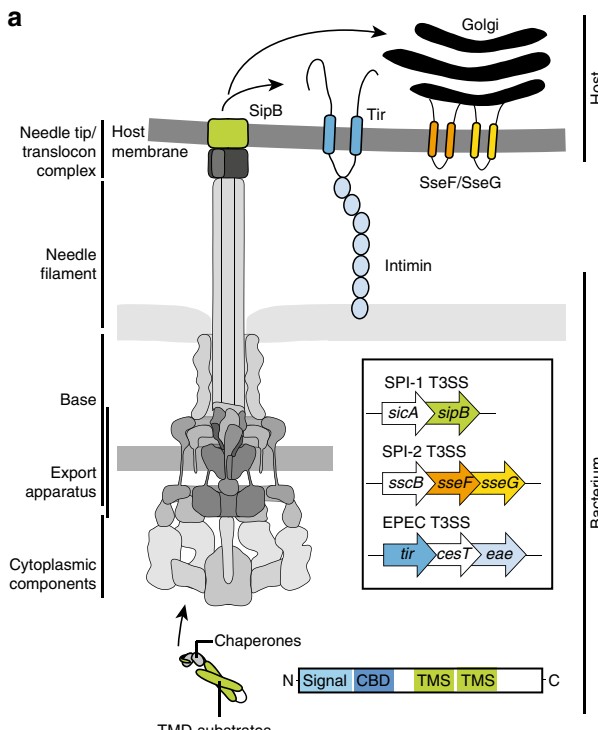

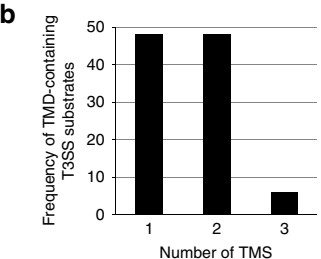

**Fig. 1** Type III-secreted transmembrane proteins **a** Type III-secreted transmembrane proteins harbor an N-terminal type III secretion signal and a chaperone binding domain (CBD). They are targeted by cognate chaperones to the sorting platform (cytoplasmic components) of the T3SS injectisome. They are secreted into the host cell and incorporated into the host membrane. SipB (green) is a hydrophobic translocator of the SPI-1 T3SS of *Salmonella*, forming pores for T3SS effector translocation. SipB is targeted by the chaperone SicA. SseF (orange) and SseG (green) are effector proteins of the SPI-2 T3SS and targeted by the chaperone SscB. Both proteins interact with endosomal compartments e.g., the Golgi network. The Tir protein (blue) of EPEC is secreted through the T3SS and inserted into the host membrane in order to act as receptor for the adhesin intimin. Tir is targeted by the chaperone CesT. **b** T3SS substrates were analyzed using the full protein scan of the ΔG predictor (window: 18–35 aa, length correction: OFF). The histogram shows the distribution of the number of TMS of type III-secreted transmembrane proteins. CBD chaperone binding domain, IM inner membrane, OM outer membrane, SPI-1 *Salmonella* pathogenicity island 1, SPI-2 *Salmonella* pathogenicity island 2, TMD transmembrane domain, TMS transmembrane segment

Given the intermediate $\Delta G_{app}$ of type III-secreted transmembrane proteins, we sought to experimentally validate the propensity of membrane integration of their first predicted TMS in order to clarify whether these predicted membrane proteins would classify as *bona fide* integral membrane proteins in a natural membrane setting. To this end, we adopted an *Escherichia coli* GlpG-based membrane integration assay[21]. In this assay, a test-segment (H) is inserted into a chimeric construct of *E. coli* leader peptidase LepB (H1 and H2, P2) and TMS 2 of lactose permease LacY (R) (Fig. 2b). The topological

orientation of the R-segment of the chimera depends on the membrane integration of the test-segment (Fig. 2b). In case of an inverted topology of the R-segment (C-in) caused by the unsuccessful membrane integration of the H-segment, the intramembrane protease GlpG cleaves the R-segment and a fragment of a smaller size is released. The relative membrane integration is calculated from the ratio of uncleaved to cleaved chimera observed by SDS PAGE and Western blotting of whole cell lysates and subsequent immunodetection of the chimera's P2-domain.

Ala-Leu test-segments of increasing hydrophobicity were used as technical controls and showed a strong positive correlation of membrane integration and number of leucines (Supplementary Fig. 1) as published previously[21]. We then tested the relative integration of the first TMS of the *Salmonella* SPI-1 translocator protein SipB, the *Salmonella* SPI-2 substrate SseF, as well as the EPEC translocated intimin receptor Tir (Supplementary Fig. 2). Because of an ambiguous prediction and the extensive length of the predicted first TMS of SipB and SseF, we chose to test two segments of different extent in each case. The relative integration of the tested TMS reached 75% for SipB$_{320–353}$, SseF$_{64–85}$, and SseF$_{86–104}$, 67% for SipB$_{320–337}$, and 60% for Tir$_{234–254}$, clearly indicating that the tested TMS can promote membrane insertion of the respective proteins to a substantial degree (Fig. 2c). In summary, our data show that T3SS TMD-substrates are *bona fide* TM-proteins albeit with a lower hydrophobicity than single-span TMD-proteins.

**T3SS TMD-substrates lack inner membrane targeting signal.** The observed intermediate hydrophobicity suggests that type III-secreted transmembrane proteins may have evolved to readily integrate into membranes of host cells while at the same time avoid classical membrane targeting pathways inside the bacterium that rely on strongly hydrophobic targeting signals[17].

On the example of the *Salmonella* T3SS substrates SipB and SseF, we sought to investigate the efficiency of discrimination between targeting to the cytoplasmic membrane and type III secretion. Analysis by SDS PAGE, Western blotting, and immunodetection showed that both proteins, SipB and SseF, were secreted into the culture supernatant in the presence of a functional T3SS but resided inside bacteria in type III secretion-deficient mutants (Supplementary Fig. 3a). To see, how much SipB and SseF, respectively, might be erroneously integrated into the bacterial cytoplasmic membrane, we fractionated *Salmonella* into inner and outer membranes using sucrose gradient centrifugation as reported previously[22]. While SipB was readily detected in Western blots of whole cell lysates (Supplementary Fig. 3a), it was not detected in inner or outer membrane fractions, neither in the wild type nor in a type III secretion-deficient strain lacking the major export apparatus protein InvA (Supplementary Fig. 3b). In contrast, SseF was found in fractions of lower sucrose density. However, as previously observed for injectisome components[8], SseF's fractionation pattern was not completely congruent with the inner membrane marker protein YidC (Supplementary Fig. 4a, b). To distinguish membrane integrated SseF from SseF that is only peripherally membrane associated, e.g., by association with the sorting platform of the injectisome, we performed protein extraction with 8 M urea. While the integral membrane protein YidC remained within the membranes, SseF was entirely extracted upon urea treatment (Supplementary Fig. 4c), indicating that it was not integral to the membrane. In summary, our observations demonstrate that both tested type III-secreted transmembrane proteins were not erroneously integrated into the bacterial inner membrane but were specifically secreted out of the cell in a T3SS-dependent manner.

Based on the above-presented results, we reasoned that the TMS of type III-secreted transmembrane proteins might not be sufficiently hydrophobic to serve as signals for inner membrane targeting and

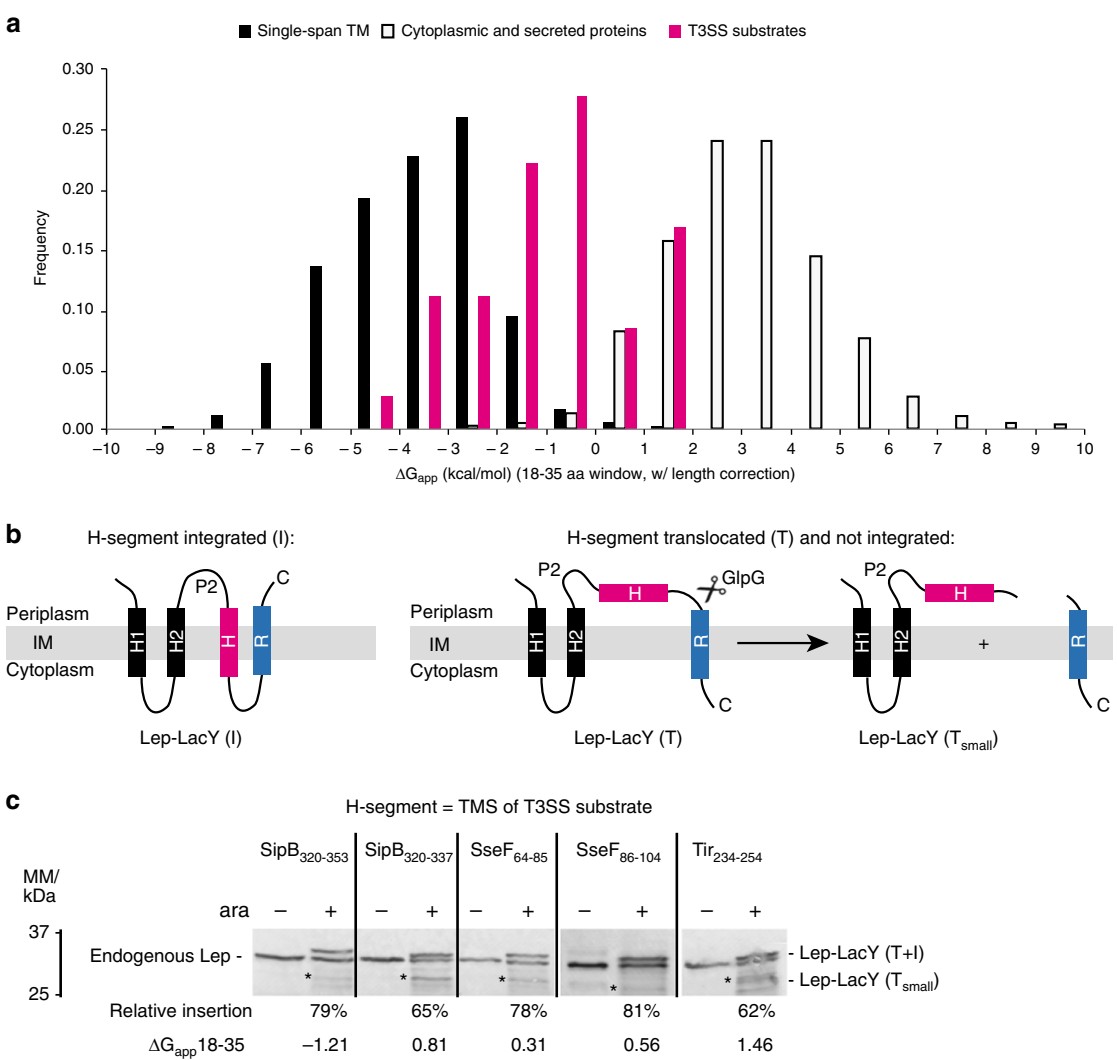

**Fig. 2** Prediction and experimental validation of membrane integration of TMS of T3SS substrates **a** Distribution of the calculated membrane integration propensity ($\Delta G_{app}$) of TMS of T3SS substrates (red) compared to previously published values for $\Delta G_{app}$ of regular transmembrane- and soluble proteins, respectively[20]. For each protein, only its lowest $\Delta G_{app}$ of any given sequence window is shown ($\Delta G$ predictor settings: window size: 18–35 aa, length correction: ON). **b** Principle of the TMS insertion assay, see results and methods sections for description[21]. **c** Relative membrane insertion of the indicated TMS was analyzed in *E. coli* BW25113 with the TMS insertion assay. ± ara indicates induction of expression of the respective test construct. The fate of the Lep-LacY chimera was assessed by SDS PAGE, Western blotting and immunodetection of the Lep P2 domain. A representative result of three independent experiments is shown. The fraction of inserted Lep-LacY ($f_I$) was calculated from the ratio of uncleaved to cleaved (indicated by asterisk) Lep-LacY, corrected for different degradation rates of the individual fragments according to Öjemalm et al[21].. Abbreviations: ara: arabinose; Lep: leader peptidase; Lep-LacY (T+I): translocated and integrated membrane helix; Lep-LacY (T_small): small translocated form of Lep-LacY. IM inner membrane, TMS transmembrane segment

thus the intermediate hydrophobicity may provide the key to pathway discrimination. Schibich et al. recently showed that targeting by the *E. coli* SRP requires a hydrophobic signal of 12–17 amino acids that corresponds to a $\Delta G_{app}$ of the targeted TMS (window 19–23 aa) of <0 kcal/mol[17]. We therefore employed the $\Delta G_{app}$ calculation to estimate the potential of a given polypeptide for SRP-dependent targeting using a window of 12–17 aa without length correction. Using these settings, reported SRP substrates[17] peaked at a $\Delta G_{app}^{12–17}$ −2.0 kcal/mol, TMS of SRP-independent membrane proteins peaked at −1.0 kcal/mol, and N-terminal TMS skipped by SRP of otherwise SRP-targeted proteins peaked at −0.5 kcal/mol (Fig. 3a). The amino acid segments of lowest $\Delta G_{app}^{12–17}$ of type III-secreted transmembrane proteins were most frequent between $\Delta G_{app}^{12–17}$ −1.5 and 0.5 kcal/mol (Fig. 3a). These data suggest that most type III-secreted transmembrane proteins behave like SRP-independent membrane proteins and skipped TMS of SRP

substrates and are not recognized by SRP for subsequent membrane targeting because of an insufficient hydrophobicity.

We sought to experimentally validate this hypothesis by testing the membrane targeting potential of the relevant TMS of SipB, SseF, and Tir (SipB_320–353, SipB_320–337, SseF_64–85, SseF_86–104, and Tir_234–254) in two well-established S-35 Met-based pulse-chase targeting assays[23] using inverted leader-peptidase (Lep-inv) and a ProW Nt/TM1/P2-chimera as vehicles. Both assays rely on the proteinase K-dependent degradation of periplasm-exposed protein domains after productive inner membrane targeting and integration (Fig. 3b, c). The accessible outer membrane protein OmpA served as a digestion control in this assay. While the wild type Lep-inv and ProW Nt/TM1/P2 proteins were readily cleaved by proteinase K as judged by immunodetection, demonstrating their efficient inner membrane targeting and integration, the respective chimeras containing the TMS of SipB_320–353, SipB_320–337, SseF_64–85, or Tir_234–254 did not seem to be targeted to and

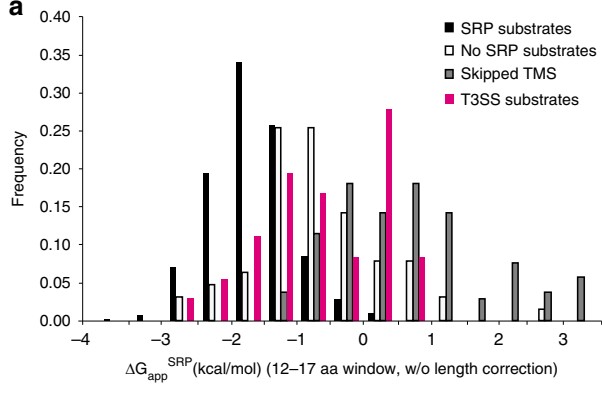

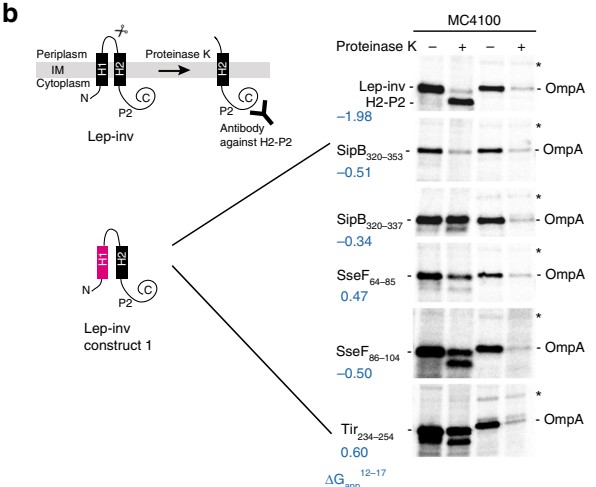

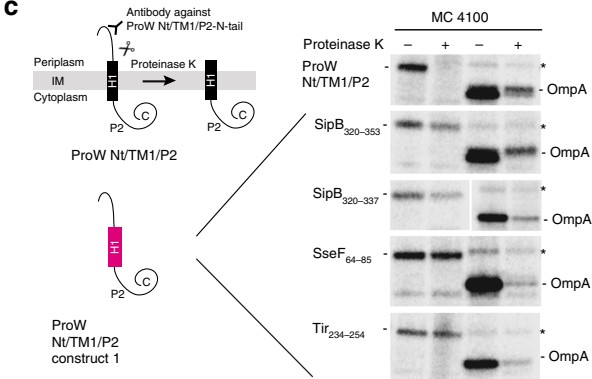

**Fig. 3** Membrane targeting potential of TMS of type III-secreted transmembrane proteins (**a**) Calculation of $\Delta G_{app}$ for the SRP-targeting window of 12–17 aa ($\Delta G_{app}^{SRP}$) for transmembrane proteins of type III-secreted transmembrane proteins (red) compared to *E. coli* transmembrane proteins. The classification of *E. coli* membrane proteins is according to Schibich et al.:[17] SRP substrates (dark gray), non SRP substrates (middle gray) and substrates, in which the first TMS was skipped by SRP (light gray). **b**, **c** Relevant TMS of T3SS substrates were assessed for their inner membrane targeting potential and membrane integration in a S-35 Met-based pulse-chase targeting assay using inverted leader-peptidase (Lep-inv, **b**) and ProW Nt/TM1/P2 (**c**), respectively[23]. The principle of the assays is shown on the left. Membrane-inserted Lep-inv/ProW Nt/TM1/P2 were cleaved into a smaller fragment by exogenously added proteinase K. Lep-inv/ProW Nt/TM1/P2 that fails to insert into the membrane is not affected by proteinase K. Cleavage can be detected by immunoprecipitation. For assessment of targeting, the H1 segment of Lep-inv or ProW Nt/TM1/P2, respectively, was exchanged against the indicated segment of interest. Proteins were expressed in *E. coli* MC4100 from rhamnose-inducible plasmids. After spheroplasting and addition of proteinase K, proteins of interest were immunoprecipitated and analyzed by SDS PAGE and autoradiography of 35-S. The outer membrane protein OmpA served as control for successful proteinase K digestion. Band X (indicated by asterisk) was used as a control for intact spheroplasts. A representative result of three independent experiments is shown. IM inner membrane, TMS transmembrane segment

Transmembrane proteins are not only secreted through T3SS but also through T4BSS of e.g., *Legionella* and *Coxiella spp*[12,24]. In order to evaluate whether the hydrophobicity niche that we observed for type III-secreted transmembrane proteins had also evolved for T4BSS and thus represented a more general solution to the problem of targeting discrimination of secreted transmembrane proteins, we investigated type IVB-secreted transmembrane proteins by the same means as the ones of T3SS. We identified 82 type IVB-secreted transmembrane proteins out of the 280 reported T4BSS substrates of *Legionella pneumophila* Philadelphia and 16 out of 127 of *Coxiella burnetii* Nine Mile strain RSA 493 (Supplementary Fig. 8). The minimal $\Delta G_{app}^{18-35}$ of these secreted transmembrane proteins was most frequent between −3 and 2 kcal/mol, similar to T3SS TM-substrates and clearly distinct from single-span transmembrane proteins (Supplementary Fig. 9). This intermediate hydrophobicity was also reflected in the SRP-targeting relevant $\Delta G_{app}^{12-17}$: Again, type IVB-secreted transmembrane proteins grouped with type III-secreted transmembrane proteins with a minimal $\Delta G_{app}^{12-17}$ most frequent between −1.5 and 0.5 kcal/mol (Supplementary Fig. 10). This observation was corroborated by data of the Lep-inv targeting assay that showed no proteinase K degradation for three of the four tested TMS of type IVB-secreted transmembrane proteins (Supplementary Fig. 11). A fourth substrate with a TMS of LegC3 showed, similar to Lep-inv-SseF$_{86-104}$, a substantial susceptibility to proteinase K leading to about 50% degradation, which indicates partial membrane integration.

Altogether, the data presented on type III-secreted and type IVB-secreted transmembrane proteins suggest that fine-tuning of hydrophobicity represents a more general solution to the problem of targeting discrimination of membrane proteins.

**The chaperone SscB aids in targeting discrimination of SseF.** While the intermediate hydrophobicity observed among T3SS and T4BSS TMD-substrates provides a passive discrimination between secretion pathways, our data for SseF$_{86-104}$ indicate that additional, possibly active mechanisms for avoidance of erroneous targeting exist.

integrated into the membrane (Fig. 3b, c, Supplementary Fig. 5). In contrast to the other tested chimeras, about half of Lep-inv-SseF$_{86-104}$ was cleaved by proteinase K, suggesting that SseF$_{86-104}$ can mediate membrane targeting and integration to a substantial degree (Fig. 3b, Supplementary Fig. 5). In particular Lep-inv-SipB$_{320-353}$ showed a substantial decrease in the overall signal upon proteinase K treatment, which might be the consequence of erroneous post-translational targeting of this chimera to the periplasm.

Together, the experimental results corroborate the computational analysis and suggest that most type III-secreted transmembrane proteins occupy an intermediate hydrophobicity niche that prevents their futile targeting to the bacterial inner membrane and allows for T3SS-dependent secretion. In agreement with this notion we observed that an increase in hydrophobicity of the first TMS of SipB and SseF, respectively, resulted in a reduced secretion of these proteins into the culture supernatant, even though the reduction in secretion did not strictly follow the increase in hydrophobicity (Supplementary Figs. 6, 7).

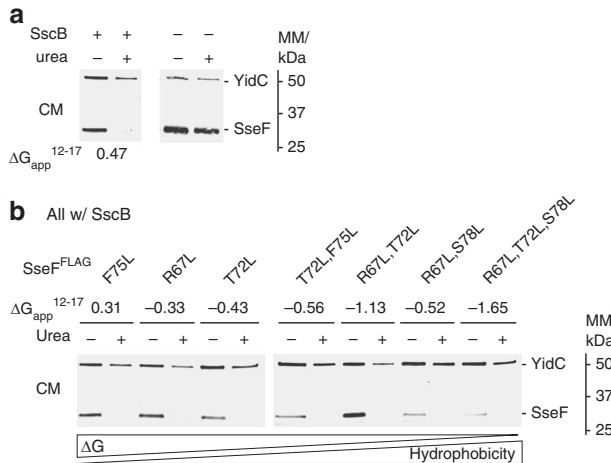

**Fig. 4** Analysis of membrane integration of SseF in dependence of its chaperone SscB (**a**) *Salmonella* ΔsscB, ΔsseF, ΔsseG triple mutants were grown under SPI-2-inducing conditions and complemented with SseF[FLAG] with or without the chaperone SscB from a rhamnose-inducible low-copy number plasmid. Crude membranes were prepared and treated with 8 M urea for 1 h at room temperature as indicated. SseF content was then analyzed by SDS PAGE, Western blotting and immunodetection with anti-FLAG and anti-YidC (inner membrane control) antibodies. **b** As in **a** but showing expression of indicated SseF[FLAG] mutants with increased hydrophobicity of the first TMS. A representative result of three independent experiments is shown. Abbreviation: CM: crude membranes

T3SS-substrates are usually targeted to the system's cytoplasmic sorting platform by cognate chaperones. Given the substantial hydrophobicity of SseF$_{86-104}$ and its ability to target Lep-inv-SseF$_{86-104}$ to the inner membrane, we hypothesized that co-translational binding of the chaperone SscB to the N-terminal CBD of its substrate SseF may prevent recruitment of SRP to the further downstream-localized hydrophobic TMS and by this actively support secretion pathway discrimination. To test this hypothesis, we assessed SseF's dependence on SscB for avoiding membrane integration using urea extraction of crude membrane preparations followed by SDS PAGE, Western blotting, and immunodetection of the FLAG-tagged SseF. As observed in the sucrose gradient centrifugation experiment, SseF did not integrate into membranes in the presence of SscB (Fig. 4a, Supplementary Fig. 12). However, in the absence of SscB, SseF readily remained associated with the membranes upon protein extraction by 8 M urea (Fig. 4a, Supplementary Fig. 12). In contrast to the peripherally membrane associated SseF observed upon SscB-SseF co-expression (Supplementary Fig. 4), SseF did co-purify with inner membrane fractions upon sucrose gradient centrifugation when expressed without its cognate chaperone (Supplementary Fig. 13). Both results clearly point towards an active role of SscB to prevent erroneous membrane targeting and integration. Even an increase in hydrophobicity of the first predicted TMS of SseF well beyond the SRP-targeting threshold did not result in erroneous membrane targeting in the presence of SscB, showing the power of this chaperone to support targeting discrimination (Fig. 4b, Supplementary Fig. 14).

**SscB directly binds to the TMD of SseF**. Cognate chaperones of T3SS were shown to specifically bind the N-terminal CBD of T3SS substrates[25]. We reasoned that prevention of erroneous membrane targeting by SscB might not only involve binding to the CBD but also to the TMD of SseF. To test this, we employed an in vivo photocrosslinking approach based on the genetically encoded UV-reactive amino acid *para*-benzophenylalanine

(pBpa)[26]. pBpa was built into the CBD (at A24) and into the first predicted TMS (L69, V73, V83, L95) of SseF, respectively (Fig. 5a). All SseF-pBpa mutants were successfully secreted into the culture supernatant in a T3SS-dependent manner, although to varying degrees (Fig. 5b, Supplementary Fig. 15c). Crosslinking of pBpa to nearby interaction partners was induced by UV irradiation of intact bacterial cells immediately after harvesting. Subsequently, crosslinking patterns were analyzed by SDS PAGE of whole cell lysates followed by immunodetection of the FLAG-tagged SseF. As expected, a crosslinking-specific band of 43 kDa that could correspond to crosslinked SseF-SscB was identified with pBpa at a position within the CBD (A24) (Fig. 5b, Supplementary Fig. 15a). However, also pBpa at positions within the first TMS (L69 and V73) showed strong crosslinks resulting in the same 43 kDa band. To a lesser extent, this was also the case when crosslinking at positions V83 and L95. The crosslinked products were not observed when expressing SseF in the absence of its cognate chaperone SscB (Fig. 5c, Supplementary Fig. 15b, c). Mass spectrometrical analysis of a Coomassie-stained gel slice cut at the position of the crosslinked product with or without pBpa at position V73 yielded the pBpa-specific identification of SscB (Supplementary Fig. 16, Supplementary Data 2). Both results provide evidence that the 43 kDa band corresponded to crosslinked SseF-SscB. Binding of SscB to the first TMS of SseF was also observed in SseF mutants with increased TMS hydrophobicity (Fig. 5d, Supplementary Fig. 15d, e).

Together, the presented data show that binding of the T3SS chaperone SscB not only to the CBD but also to the TMS of its cargo SseF efficiently prevents erroneous inner membrane targeting and insertion of this type III-secreted transmembrane protein, even when the hydrophobicity of its first TMS is increased well beyond the SRP-targeting threshold.

## Discussion

The ability to secrete transmembrane proteins is essential for the function of T3SS and T4BSS and for the pathogenicity of bacteria utilizing these systems. Despite the high biological relevance of the secretion of these substrates, it had not been investigated how transmembrane proteins can be secreted through T3SS and T4BSS and avoid erroneous targeting to and integration into the bacterial inner membrane. Here, we provide evidence for a highly effective two-pronged mechanism of substrate discrimination that prevents erroneous inner membrane targeting of type III-secreted transmembrane proteins even in the absence of active type III secretion: First, a balanced hydrophobicity of TMS of most type III-secreted transmembrane proteins prevents targeting to the bacterial inner membrane but permits membrane integration as required in the host. This mechanism of passive targeting-avoidance may be a general principle, which we also identified for a subset of substrates of T4BSS of *Legionella* and *Coxiella*. Second, erroneous inner membrane targeting of more hydrophobic type III-secreted transmembrane proteins is actively prevented by binding of their cognate chaperone not only to their CBD but also to their TMS, which may impede recruitment of other targeting factors such as SRP (Fig. 6).

Integration of polypeptides into biological membranes requires the presence of a stretch of sufficiently hydrophobic amino acids that corresponds to the physico-chemical properties of the cross-section of the lipid bilayer and thus yields a negative ΔG$_{app}$ upon membrane integration[20,27]. Targeting factors such as SRP have evolved to recognize these hydrophobic sequences as signals to guide transmembrane proteins to the target membrane. The correlation between the ΔG$_{app}$ of membrane integration of a given TMS and its propensity for targeting by SRP have recently been worked out on a global level in *E. coli*[17]. The study

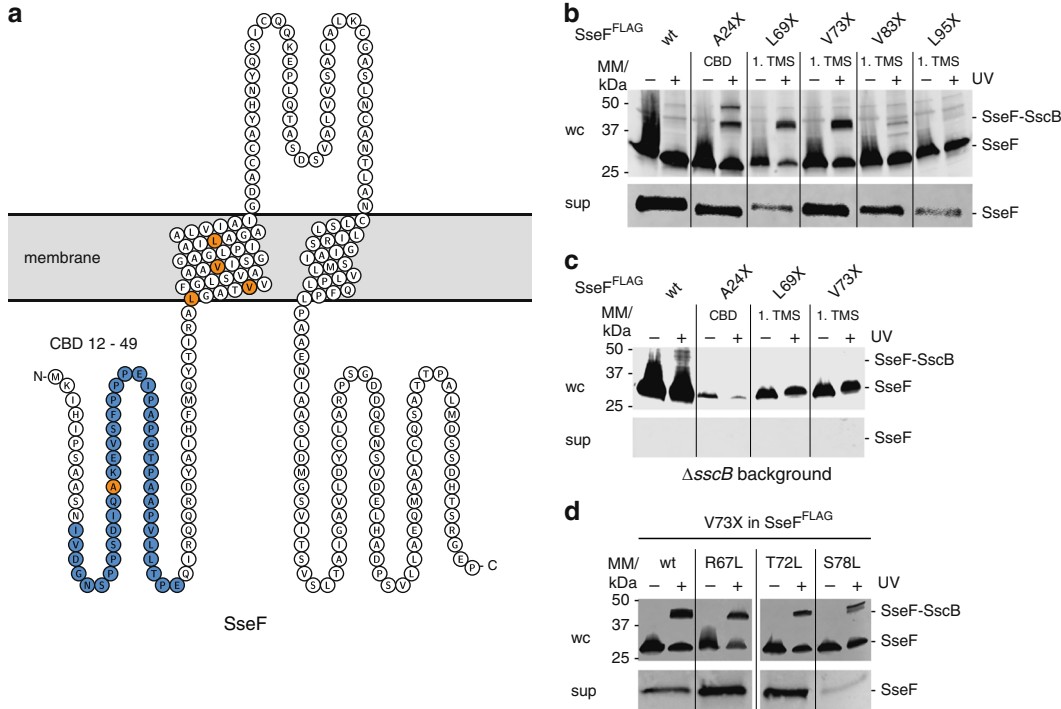

**Fig. 5** Protein-protein interaction analysis of SseF and SscB by in vivo photocrosslinking **a** Protter[39] visualization of SseF presenting predicted TM topology, chaperone binding domain (blue), and positions analyzed by in vivo photocrosslinking (orange). **b** Immunodetection of SseF[FLAG] in whole cell lysates with and without UV irradiation (upper panel). pBpa mutations are denoted as "X". T3SS-dependent secretion of mutated SseF was analyzed by SDS PAGE, Western blotting and immunodetection of TCA-precipitated SseF[FLAG] in culture supernatants (lower panel). **c** As in **b** but showing SseF[FLAG] in a ΔsscB deletion background. **d** As in **b** but showing crosslinking position V73 of SseF in mutants harboring various leucine residues on the indicated positions of the first TMS of SseF. A representative result of three independent experiments is shown. CBD chaperone binding domain, WC whole cell lysates, sup culture supernatant, TMS transmembrane segment

uncovered that SRP-targeted TMS exhibit a lower $\Delta G_{app}$ for sequence windows of 19–23 aa than needed for simple membrane integration and we showed herein that the $\Delta G_{app}$ for the actual SRP-targeting window of 12–17 aa is even more distinct. This difference in $\Delta G$ between SRP targeting and membrane integration reflects that SRP can only bind polypeptides of a limited length of 12–17 aa while membranes can accommodate TMS of a much wider range (18–40 aa) and can maximize beneficial interactions by tilting of the helices. Consequently, SRP-binding polypeptides are required to exhibit a high hydrophobic density while TMS may be of lower hydrophobic density given a sufficient length that compensates this deficit. The herein presented data indicate that type III-dependent and type IVB-dependent secretion of transmembrane proteins is enabled by exploiting the difference in hydrophobic density that is required for membrane integration and SRP-targeting. We observed that TMS of type III-secreted transmembrane proteins tend to be longer and exhibit a lower hydrophobic density, sufficient to promote membrane integration but predicted to be insufficient for recognition by SRP. The same general mechanism seems to apply for the secretion of many transmembrane proteins through T4BSS.

Since TMD-effectors enter the host cell through injection, membrane targeting and insertion of these proteins are—unlike for ribosomally made host proteins—post-translational processes. While evidence suggests that membrane insertion of translocon-type TMD-effectors is aided by the needle tip of the injectisome[28,29], it is not clear at this point whether host proteins aid in membrane targeting and insertion of non-translocon type TMD-effectors. It is striking that most investigated type III-secreted TMD-effectors insert into the membrane that is directly targeted by the T3SS injectisome (e.g., Tir, SseF, SseG), however, the mechanism of membrane insertion may not involve lateral

partitioning through the T3SS translocon as recent data for Tir suggest[30] but cytoplasmic intermediates[31]. Hence, in addition to playing a role in targeting discrimination, the intermediate hydrophobic density of TMS of TMD-effector proteins may also serve to prevent aggregation of these proteins in the host cell cytoplasm while being sufficient to mediate association of their TMD with the interface of lipid bilayers of host cells and subsequent membrane insertion. Membrane insertion may also be facilitated by the mostly low complexity of the TMDs of secreted transmembrane proteins (Fig. 1b, Supplementary Fig. 8).

While the passive targeting discrimination by a reduced hydrophobic density may be sufficient for most T3SS substrates, we observed that substrates with TMS of stronger hydrophobicity, like SseF, also require the action of cognate T3SS chaperones to avoid erroneous targeting to the bacterial inner membrane. This mechanism necessitates co-translational binding of the T3SS chaperone to the N-terminal CBD and to the TMS of type III-secreted transmembrane proteins in order to overrule the in general co-translational inner membrane targeting by factors like SRP. In consequence, our data suggest that T3SS merge co-translational targeting of substrates with post-translational secretion. The sorting platform resembles the injectisome's targeting receptor. Recent studies showed that sorting platform components are in dynamic exchange between an injectisome-bound and a free cytoplasmic form[7,32–34] and the latter may be the target of chaperone-substrate complexes. The resulting tri-partite complex may thus represent the link between co-translational targeting and post-translational secretion and may be the storage form of substrates in charged bacteria that awaits secretion and injection upon host cell contact.

Intracellular storage of type III-secreted transmembrane proteins requires the shielding of the hydrophobic TMS to prevent

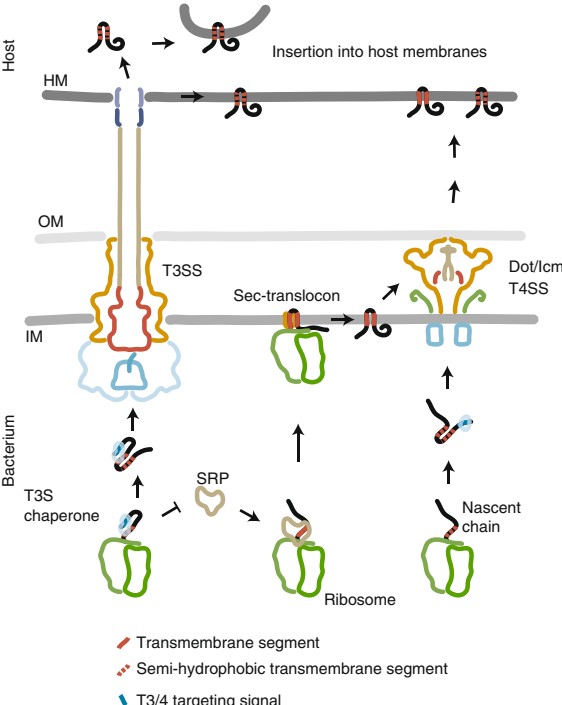

**Fig. 6** Model of targeting discrimination of secreted transmembrane proteins The semi-hydrophobic nature of most TMS of type III-secreted and type IVB-secreted transmembrane proteins prevents recruitment of SRP and erroneous targeting to the Sec-translocon. In addition, specific T3SS chaperones bind TMS of type III-secreted transmembrane proteins and thus shield these segments from wrong targeting factors and aggregation. T4BSS can also accept strongly hydrophobic substrates from within the bacterial inner membrane. HM host membrane, IM inner membrane, OM outer membrane

erroneous targeting or aggregation. While general chaperones may be involved in protecting these T3SS substrates, the herein observed binding of the cognate chaperone SscB not only to SseF's CBD but also to its TMS suggests that the T3SS chaperones also play an important role in this process. This view is supported by the previously made observation of the co-crystal structure of the *Aeromonas* translocator AopB bound to its chaperone AcrH, that also showed a binding of the TMS of AopB to the chaperone[35]. While the authors speculated that the observed interaction reflects the mechanism of insertion of the translocator into the host cell membrane, our data rather point at the eminent role of this interaction in targeting and protection of these hydrophobic substrates. Along the same lines, shielding of membrane localization domains that serve in targeting some T3SS effectors to the periphery of host membranes was also reported to involve engagement of cognate T3SS chaperones[36]. The comparably low hydrophobicity of these domains would not result in membrane targeting inside the bacterium but the cognate chaperones were shown to prevent intrabacterial aggregation of the respective effectors.

Our data on SseF show that several factors may contribute to targeting discrimination. SseF's cognate chaperone SscB prevented futile membrane targeting also when SseF contained TMS of strong hydrophobicity (Fig. 4b), however, the stronger hydrophobicity did still reduce secretion (Supplementary Fig. 6d). In particular for SseF mutant S78L it is conceivable that reduced secretion is a consequence of improper chaperone binding. In fact, we did observe an altered crosslinking profile between SseF$_{S78L}$ and SscB (Fig. 5d). We also noticed a reduced accumulation of this mutant at the membrane (Fig. 4b), which may be

the consequence of impaired targeting to the sorting platform of the injectisome, caused by improper chaperone binding.

In contrast to substrates of T3SS, substrates of T4BSS harbor C-terminal secretion signals and ill-defined binding sites for the system's chaperones, consequently, co-translational mechanisms of targeting discrimination cannot play a role. We show here that many type IVB-secreted transmembrane proteins possess TMS of low hydrophobic density and it is comprehensible that this passive mechanism suffices for efficient targeting discrimination. Also in contrast to T3SS, however, T4BSS have evolved to accept some substrates from within the bacterial inner membrane, even though the underlying secretion mechanism is unknown. Extraction of transmembrane proteins from the lipid bilayer is a highly energy-consuming process, which may explain why secretion of transmembrane proteins of lower hydrophobic density is the preferred mechanism also in T4BSS.

In summary, our study highlights the delicate mechanisms that have evolved inside bacteria to enable secretion of transmembrane proteins that act in various and often essential capacities to promote bacteria-host interactions of numerous pathogens and symbionts.

## Methods

**Materials**. Chemicals were from Sigma-Aldrich unless otherwise specified, para-benzophenylalanine was from Bachem. The radioactive [S35]-methionine was obtained from Hartman Analytics. SERVAGel™ TG PRiME™ 8–16% precast gels were from Serva. Primers are listed in Supplementary Data 3 and were synthetized by Eurofins and Integrated DNA Technologies. Monoclonal M2 anti-FLAG antibody was from Sigma-Aldrich (F3165). Secondary antibodies goat anti-mouse IgG DyLight 800 conjugate and goat anti-rabbit IgG DyLight 680 conjugate were from Thermo-Fisher (SA5-35571, 35568, respectively).

**Bacterial strains and plasmids**. Bacterial strains and plasmids used in this study are listed in Supplementary Data 3. Primers for construction of strains and plasmids were listed in Supplementary Data 3. All *Salmonella* strains were derived from *S.* Typhimurium strain SL1344[37]. Bacterial cultures were supplemented as required with streptomycin (50 μg/mL), tetracycline (12.5 μg/mL), ampicillin (100 μg/mL), kanamycin (25 μg/mL), or chloramphenicol (10 μg/mL). Molecular cloning was performed by standard Gibson cloning using templates and primers as listed in Supplementary Data 3[38]. Site-directed mutagenesis was performed following the Quik Change protocol (Stratagene) using KOD (Novagen) or Phusion polymerase (Thermo) using templates and primers as listed in Supplementary Data 3.

**Prediction of transmembrane segments**. The position and length of TMS was predicted using dGpred in the full protein scan mode[20]. Windows of 18–35 amino acids were analyzed with amphiphilicity and length correction. With reference to Öjemalm et al.[21], only predicted TMS with ΔG < 1.5 kcal/mol were considered. For evaluation of the SRP-targeting potential, dGpred full protein scan was used with analysis windows of 12–17 aa without regarding length correction. For visualization, the online tool PROTTER was used[39].

**Secretion assay**. Analysis of type III-dependent secretion of proteins into the culture medium through the T3SS encoded by SPI-1 was carried out as described previously[40]. *S.* Typhimurium strains were grown at 37 °C in LB broth supplemented with 0.3 M NaCl for 5 h. Whole cells and culture supernatants were separated by centrifugation for 2 min at 10,000×g. Whole cells were resuspended in 75 μl SDS PAGE loading buffer[41]. Supernatants were passed through a 0.2 μm filter, supplemented with 0.1% Na-deoxycholic acid, and precipitated with 10% trichloroacetic acid (TCA) for 30 min on ice. Precipitated culture supernatant samples were washed with acetone after centrifugation for 30 min at 20,000×g and 4 °C and resuspended in 40 μl SDS PAGE loading buffer. Fifteen microliter of whole-cell and 20 μl of culture supernatant samples were subjected to SDS PAGE, followed by immunoblotting.

To analyze type III-dependent secretion of proteins into the culture medium through the SPI-2 encoded T3SS, *S.* Typhimurium strains were transformed with a constitutively active pLacwoI plasmid harboring a truncated H-NS version to enhance SPI-2 gene expression[42]. Cells were grown at 37 °C for 5 h in LB broth supplemented with 0.3 M NaCl and additionally 80 mM MES pH 5.8 to induce SPI-2 gene expression[43].

**Immunoblotting**. For protein detection, samples were subjected to SDS PAGE using SERVAGel™ TG PRiME™ 8–16% precast gels, transferred onto a PVDF membrane (Bio-Rad), and probed with primary antibodies anti-Lep[44] (1:10,000), anti-SipB[40] (1:1000), anti-Inv[40] (1:5000), anti-YidC[45] (1:10,000), anti-OmpA[46]

(1:20,000), anti-ProW$_N$[47] (1:10,000), and M2 anti-FLAG (1:10,000). Secondary antibodies were goat anti-mouse IgG DyLight 800 conjugate (1:10,000) and goat anti-rabbit IgG DyLight 680 (1:10,000). Scanning of the PVDF membrane and image analysis was performed with a Li-Cor Odyssey system and image Studio 2.1.10 (Li-Cor).

**In vivo photocrosslinking**. S. Typhimurium ΔsscB, pMIB5873 was grown at 37 °C in LB Lennox. pMIB5873 constitutively expressed ssrB and a truncated allele of hns (H-NS Q92am) to enhance expression of SPI-2. Cultures were supplemented with 1 mM rhamnose to induce expression of SseF$^{FLAG}$ from low copy number pTACO10 plasmids[8] and with the artificial amino acid para-benzoyl phenyl alanine (pBpa, final concentration 1 mM). Cultures were incubated for 5.5 h at 37 °C and 200 r.p.m. Two ODU of bacterial cells were harvested and washed once with 1 mL cold PBS. Cells are resuspended in 1 mL PBS and transferred into 6-well cell culture dishes. Bacteria were irradiated at λ = 365 nm on a UV transilluminator table (UVP) for 30 min.

**Crude membrane preparation**. Crude membranes were prepared as reported previously[9]. 10 OD units of S. Typhimurium cultures were resuspended in 750 μl buffer K (50 mM triethanolamine (TEA), pH 7.5, 250 mM sucrose, 1 mM EDTA, 1 mM MgCl₂, 10 μg/mL DNAse, 2 mg/mL lysozyme, 1:100 protease inhibitor cocktail). After incubation on ice for 30 min, samples were bead milled. Beads, unbroken cells and debris were removed by centrifugation for 10 min at 10,000×g and 4 °C. Crude membranes were pelleted by centrifugation for 45 min at 55,000 r.p. m. and 4 °C in a Beckman TLA 55 rotor. Pellets containing crude membranes were frozen until use.

**Urea extraction**. Membrane samples were solubilized in 8 M urea in 2× buffer M (100 mM TEA, pH 7.5, 2 mM EDTA) for 1 h at room temperature and centrifuged for 1.5 h at 23 °C in the Beckman TLA-55 rotor at 50,000 r.p.m. Pellets were resuspended in SB loading buffer, heated at 50 °C for 15 min, and analyzed by SDS PAGE, Western blotting, and immunodetection using anti FLAG M2 antibody.

**Membrane fractionation**. Inner and outer membranes were fractionated from 1 L culture of S. Typhimurium as described previously in detail[22] using a continuous sucrose gradient (30–53% w/w). In brief, 1 L of a culture of S. Typhimurium was lysed in buffer K by two cycles of French pressing. Crude membranes were precipitated at 235,000 × g for 1 h, resuspended in buffer M (50 mM TEA, pH 7.5, 1 mM EDTA), and loaded on top of a 30–53% (w/w) continuous sucrose gradient, which was made using a Gradient Station (Biocomp, Fredericton, NB, Canada). Inner and outer membranes were separated by centrifugation at 285,000×g for 13 h. Twelve fractions of ~1.1 ml were collected using the Gradient Station.

**Lep-inv and ProW Nt/TM1/P2 proteinase K accessibility assay**. E. coli MC4100 transformed with plasmids expressing Lep-inv or ProW Nt/TM1/P2 chimeras, respectively, were grown in M9 minimal media to an OD$_{600}$ of 0.5, after which protein expression was induced for 30 min by addition of rhamnose to a final concentration of 2 mM. The cells were labeled with 2 μl [³⁵S]-methionine (30 μCi/ml culture) for 2 min. The labeling process was stopped with 100 μL cold methionine [5 mg/ml] and the cells were collected at 10,000 × g for 2 min. For spheroplasting, the cells were resuspended in 500 μl ice cold spheroplast buffer (40% (w/v) sucrose; 33 mM Tris-HCl pH 8.0), 1 μl 0.5 M EDTA and 10 μl 0.25 mg/ ml lysozyme and incubated for 15 min on ice. The spheroplasts were split for incubation for 1 h either in the presence or absence of proteinase K (final 0.4 mg/ ml). For protease inhibition, 20 μl 20 mg/ml PMSF and 300 μl 20% TCA were added and the spheroplast solution was incubated on ice for 1 h. After that, samples were washed with acetone and immunoprecipitated using anti-Lep, anti-OmpA, and anti-ProW$_N$ antibodies (1 μl each) as described previously[23].

**Transmembrane segment insertion assay**. Analysis of insertion propensity of T3SS substrates into membranes was carried out as described previously[21] with minor modifications. 2 ml cultures of E. coli transformed with the relevant plasmids were grown at 37 °C to an OD600 = 0.4, at which protein expression was induced by the addition of arabinose to a final concentration of 0.2%. After 90 min of expression, 1 ml of the culture was harvested by centrifugation at 10,000 × g for 2 min. The bacterial pellet was resuspended in 120 μl SB loading buffer, boiled for 10 min, and 10 μl of samples were analyzed by SDS PAGE, Western blotting and immunodetection using anti-Lep antibody. Bands were quantified using a LiCor Odyssey imager and the uncorrected fraction of inserted H-segment was calculated from the ratio of full length (Lep-LacY (T+I)) and cleaved (Lep-LacY (T$_{small}$) chimeras. Correction for different degradation rates of the Lep-LacY fractions was performed based on rates and formula reported by Öjemalm et al.:[21] $f_I = 1 - 1/(0.83 + 1.405 \times (T + I)/T_{small})$.

**Mass spectrometry of crosslinked interaction partners**. The identification of crosslinked proteins by mass spectrometry was performed as described previously[9]. LC-MS/MS analyses were performed on an EasyLC 1200 coupled to a QExactive HF mass spectrometer (both Thermo Scientific) essentially as described

elsewhere[48]: The peptide mixtures were injected onto the column in HPLC solvent A (0.1% formic acid) at a flow rate of 500 nl/min and subsequently eluted with a 57 min segmented gradient of 10–33–50–90% HPLC solvent B (80% ACN in 0.1% formic acid). The flow rate was at 200 nl/min during peptide elution. Full scan was acquired in the mass range from m/z 300 to 1650 at a resolution of 120,000 followed by HCD fragmentation of the 7 most intense precursor ions. High-resolution HCD MS/MS spectra were acquired with a resolution of 60,000. The target values for the MS scan and MS/MS fragmentation were $3 \times 10^6$ and $10^5$ charges with a maximum fill time of 25 ms and 110 ms, respectively. Precursor ions were excluded from sequencing for 30 s after MS/MS. The MS data were processed with MaxQuant software suite v.1.5.2.8 essentially as described previously[49–51]. Database search was performed using the Andromeda search engine[50], which is part of MaxQuant. MS/MS spectra were searched against a target database consisting of 10,152 protein entries from S. Typhimurium and 248 commonly observed contaminants. In database search, full tryptic in addition to chymotryptic specificity (cleavage C-terminal of phenylalanine, tryptophan, tyrosine, leucine, and methionine) was required and up to five missed cleavages were allowed. Carbamidomethylation of cysteine was set as fixed modification, protein N-terminal acetylation, and oxidation of methionine were set as variable modifications. The dependent peptide algorithm was enabled. Initial precursor mass tolerance was set to 4.5 parts per million (p.p.m.) and at the fragment ion level 20 p.p.m. was set for HCD fragmentation. Peptide, protein and modification site identifications were filtered at a false discovery rate (FDR) of 0.01.

**Data availability**. Data supporting the findings of this manuscript are available from the corresponding author upon reasonable request. The mass spectrometry proteomics data have been deposited to the ProteomeXchange Consortium via the PRIDE partner repository with the dataset identifier PXD010470.

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

## Acknowledgements

We thank Gunnar von Heijne and Karin Öjemalm, Stockholm University, for providing protocols and plasmids for the TMS insertion assay. We thank Boris Macek and Mirita Franz-Wachtel from the Proteome Center Tübingen for mass spectrometric-identification of the SseF-SscB-crosslink. Work performed in the laboratory of S.W. related to this project was supported by the Ministerium für Wissenschaft, Forschung und Kunst Baden-Württemberg in the framework of the Juniorprofessurenprogramm, and by the Alexander von Humboldt Foundation in the framework of the Sofja Kovalevskaja Award endowed by the Federal Ministry of Education and Research (BMBF). L.K. was supported by a short-term fellowship of the European Molecular Biology Organisation (EMBO ASTF 349-2014).

## Author contributions

L.K. performed experiments, analyzed data, and wrote the first draft of the paper. S.M. performed experiments and analyzed data. I.G. analyzed data. T.T. performed experiments. A.L. supervised experiments. J.-W.d.G. supervised experiments and analyzed data. S.W. developed the study, analyzed data, and wrote the paper.

## Additional information

**Competing interests:** The authors declare no competing interests.

