## [Peer Review file · Nature Communications]

Reviewers' comments:

Reviewer #1 (Remarks to the Author):

The manuscript by Krampen et al. uncovers a fundamental property of TM segments in secreted effectors and translocators that are exported via type III and type IV secretion systems. This work argues that improper targeting of effectors/translocators to the bacterial inner membrane is prevented by two mechanisms: intermediate hydrophobicity of the TM segments, preventing spontaneous insertion and recognition by SRP, and masking of TM segments by interactions with export chaperones. This work should be of broad interest, not just the type III secretion crowd, since it uncovers a fundamental design principle that seems to be conserved throughout bacterial secretion system cargos.

Specific comments:

- Could the authors speculate a bit on how effectors with TM domains are inserted into the host cell plasma membrane? I don't know lipid composition off-hand, but it seems odd to me that the same principles that prevent insertion into the bacterial IM would not affect insertion into the host cell membrane. (the authors mention on line 366 that the lower hydrophobicity could facilitate their insertion into the host cell plasma membrane, what is this based on?)
- Could this property be the reason why most pore-forming translocators have to rely on chaperoning by the T3SS needle tip for insertion into the host cell membrane?
- Fig. S4 suggests that ΔG_{app} is not the only determinant that affects export (e.g. it looks like the SseF R67L/S78L mutant is more defective for export than the R67L/T72L mutant (S4D), even though the ΔG_{app} values (S4C) would predict otherwise. Clearly export is not the same as membrane insertion, but the authors should comment on this. In fact, the Fig. 4B data would argue that membrane insertion is not the reason for the drop in export. Could it have to do with the chaperone interaction?

Minor comments:

- Line 124, why list *Escherichia coli*, if all other bacteria are listed by genus name only?
- Line 132, this seems a bit trivial since, at the very least, these T3SS will be secreting translocator proteins, which have TMs
- In Fig. 3B the SseF64-85 blot seems cut off too close to the full length band, so that the truncated protein wouldn't be visible.
- Would it be possible to include the ΔG_{app} values for the various TM domains, e.g. in Fig. 2C, 3B/C, and importantly, Fig. 4B

Reviewer #2 (Remarks to the Author):

The paper entitled "Revealing the mechanisms of membrane protein export by virulence associated bacterial secretion systems" focuses on the partitioning of hydrophobic membrane localization domains (MLDs) in proteins secreted via the type 3 and type 4 secretion systems. The authors show by bioinformatics analysis that MLDs in secreted proteins have lower hydrophobicity and "lower hydrophobic density" compared to the bona fide membrane proteins in *E. coli*. In addition, using a set of biochemical assays, they test the capacity of MLDs to integrate in the inner membrane and to bind to SRP. These experiments show that isolated segments of several T3SS substrates support partial membrane integration. Chaperon binding was assessed by the site directed UV crosslinking T3SS substrates. The authors elaborate a plausible model according to which intermediate hydrophobicity and chaperon binding near or at MLDs contribute to the targeting and T3S-mediated secretion of MLD-containing proteins.

Although their idea is not novel, the authors provide some new experimental support to their

claims using several complementary approaches. I am not completely convinced by the quantitative aspects of their assessments.

Major comments:

1. The idea developed in this study is not novel, although it is presented as such. The authors fail to cite previous work and to acknowledge the large number of studies that discuss and support this idea. In 2006 the Cornelis group (Letzelter et al. EMBO 25, 3223–3233) showed that the Yersinia chaperon SycO binds to the YopO MLD in the cytosol and prevents its aggregation. Deletion of MLD renders YopO chaperon-independent for secretion and prevents its final membrane localization in a eukaryotic target cell, as shown by GFP fusions. Letzelter et al. even proposed that the chaperons have emerged originally to shield aggregation-prone MLDs, before acquiring a secondary role in targeting to the secretion apparatus.

2. The membrane integration assay developed by von Hejine is used here to compare integration of several different peptides. It is very difficult to assess the results of this assay in a quantitative manner. The band of the full-length chimera never changes the intensity and there are no proper internal controls. In the pilot assay hydrophobicity of a single protein and its variants is compared, and even there the amount of the cleavage product is much higher than the amount of full-length membrane inserted protein. Minor differences in the "ratio" and the small amount of cleavage product appear to undergo important variations. Therefore, the summary of results in the form of a quantitative graph with error bars in my view is not very meaningful.

3. The proteolysis assays in Fig. 3B. show cropped images just below the full-length band and one could not see the proteolytic fragments for SipB 320-353 or SseF 64-85 if they were there. The reduction of full-length protein level is nevertheless an indication of cleavage. How do the authors interpret the behavior of Tir segment, which shows cleavage but at the same time similar protein levels? Results of the two assays do not seem to be entirely consistent. How do the authors explain the observed differences?

The charge distribution in these segments might influence membrane orientation, has this been taken into account in the predicted topology?

4. For several protocols and experiments the authors do not provide sufficient detail (e.g. urea extraction analysis). Several others have been described in a very laconic manner.

Minor comments.

1. It would be helpful if the lines in the text were numbered continuously.

2. The authors should avoid laboratory jargon. The legend of Figure 2 mentions the Keioref strain - the strain has its name, BW25113. Also, what is a skipped TMS?

Reviewer #3 (Remarks to the Author):

Major Comments

Krampen et al report on the secretion of T3SS and T4SS membrane proteins in pathogenic bacteria and shed light on how these proteins avoid being mistargeted to the Sec translocon and inserted into the membrane. The overall results show that they do so typically by possessing a weakly hydrophobic TM segment that can integrate into the membrane but is insufficiently hydrophobic to allow targeting to SRP which would direct the protein to the Sec translocon for membrane insertion. In addition, they find that many of the T3SS membrane proteins contain cognate chaperones (that bind to the amino-terminal cytoplasmic domain of the substrate and the TM segment) that prevent targeting by SRP. They propose a two-pronged mechanism, namely low hydrophobicity of the TM segments and targeting by cognate chaperones, that explains how substrates can be accurately routed for type III secretion. While I have some concerns with some of the studies and interpretations, I think the results are quite interesting and advance the field

The authors used several different approaches to study the type 3 secretion membrane proteins. First, the authors use bioinformatics to show that out of 174 T3SS known substrates there are 37 membrane protein candidates, typically with one or two TM segments (Fig. 2A). The transmembrane segments of the T3SS generally have lower overall hydrophobicity than single *E. coli* TM membrane proteins. Several candidate TM segments (derived from the type 3 secretion substrates SipB, SseF and Tir) were tested in *E. coli* using a chimeric construct where a test segment (placed in between two known TM segments) can be examined for membrane integration using a previous established assay involving GlpG cleavage (Fig. 2B). The authors discovered each of the TM segments of the type 3 secretion substrates were membrane integrated. Second, they showed only one of these TM segments could promote insertion of inverted Lep when the T3SS TM segment was interchanged with the H1 domain of inverted Lep (Fig. 3B). Similarly, the tested TM segments of T3SS could not promote translocation of the amino-terminal domain of a ProW-P2 construct when substituted for the proW first TM segment (Fig. 3C). They suggest this is because the TM segment was insufficiently hydrophobic to promote SRP targeting.

The authors next examine the authentic type 3 secretion substrate SscF (Fig. 4 and Fig. S3) and SipB (Fig. S3) in *Salmonella* and show that the substrates are secreted into the supernatant but only when the cognate chaperone SscB is present in the cell (Fig. 4). When the cognate chaperone was not present, the membrane substrate SipB was not integrated into the membrane, while SseF was membrane integrated (Fig. S9). However, SseF was not membrane integrated (Fig. S3 D and E) when the chaperone SscB was present; presumably because it prevents SRP targeting to the Sec translocon. Finally, they show, using photocrosslinking studies, that the chaperone SscB interacts with SseF substrate since it can be crosslinked to SseF when a photoprobe is placed in either the chaperone binding domain or in the first TM domain.

In addition, the authors also examined Type 4 secretion substrates. These substrates, which are secreted, can also be a membrane protein. Similar to Type 3 secretion substrates, they observed the TM segments of type 4 secretion membrane substrates had a lower hydrophobicity than single TM *E. coli* proteins. Of the four tested TM segments of T4SS proteins, only one could function in place of the first TM segment of inverted Lep and promote membrane insertion. The authors suggest that generally the TM segment of T4SS membrane proteins cannot promote SRP targeting (although one of them could).

In conclusion, the work reveals how secretion systems can specifically secrete membrane protein substrates and avoid integrating them into the plasma membrane, prior to secretion out of the cell.

Minor Comments

1. On the membrane targeting potential of transmembrane segments of T3SS protein, the authors suggest that the amino-terminal TM segments of the type 3 secretion substrates can promote insertion of the inverted Lep only when it can initiate SRP targeting. The idea is that the TM segments fail to promote membrane insertion due to the fact that they cannot be targeted to SRP. It might be best to show the TM segments that fail to promote insertion do not bind to SRP but the one candidate that does (SecF 86-104), indeed, binds to SRP. Photocrosslinking could be used to answer this question.
2. The authors should address why they observed a significant decrease in the SipB 320-353 construct when protease is added in Fig. 3B. Is this due to lysis even though band X does not decrease? Similar concern for the proW study. There is a decrease in SipB 320-353 as well as SipB 320-337 when proteinase K is added (Fig. 3C).
3. On page 4, it is stated (sentences 292-294) that an increase in hydrophobicity of the first

predicted TMS of SseF well beyond the SRP-targeting threshold did not result in erroneous membrane targeting in the presence of SscB (Fig. 4B). On page 8, it is stated "that a lower hydrophobicity of the TM segment prevents futile SRP-pathway targeting. In support of this, an increase in hydrophobicity of the first TMS of SipB and SseF, respectively, resulted in a reduced secretion of these proteins into the culture (Fig. S4)". At first glance this seems contradictory. Please clarify what you mean. Moreover, I would recommend showing directly that when the TM segment is made more hydrophobic that the protein can be integrated into the membrane.

4. In the Discussion, line 337, I would tone this down since one of the candidate TM segments of type 4 secretion membrane protein did promote membrane targeting (LegC3) (Fig. S3). Similarly, tone down line 398, where it is stated "that type IVB-secreted transmembrane proteins possess TMS of low hydrophobic density and it is comprehensible that this passive mechanism suffices for efficient targeting discrimination". Based on the author's data, it is not clear why the T4SS containing this LegC3 TM segment avoids membrane integration. This should be discussed.

Page 12 (line 355), I would change "In consequence" to "Consequently".

Response to reviewers' comments on the manuscript by Krampen et al. entitled "Revealing the mechanism of membrane protein export by virulence-associated bacterial secretion systems."

Responses are in ***bold italic*** letters.

Reviewer #1 (Remarks to the Author):

The manuscript by Krampen et al. uncovers a fundamental property of TM segments in secreted effectors and translocators that are exported via type III and type IV secretion systems. This work argues that improper targeting of effectors/translocators to the bacterial inner membrane is prevented by two mechanisms: intermediate hydrophobicity of the TM segments, preventing spontaneous insertion and recognition by SRP, and masking of TM segments by interactions with export chaperones. This work should be of broad interest, not just the type III secretion crowd, since it uncovers a fundamental design principle that seems to be conserved throughout bacterial secretion system cargos.

We thank the reviewer for highlighting the relevance of our work.

Specific comments:

- Could the authors speculate a bit on how effectors with TM domains are inserted into the host cell plasma membrane? I don't know lipid composition off-hand, but it seems odd to me that the same principles that prevent insertion into the bacterial IM would not affect insertion into the host cell membrane. (the authors mention on line 366 that the lower hydrophobicity could facilitate their insertion into the host cell plasma membrane, what is this based on?)

Our analysis deals with the question how erroneous targeting of T3SS TMD-substrates to the bacterial inner membrane is prevented. Targeting to the bacterial inner membrane is by and large (with the exception of C-tail anchored proteins) a co-translational process, involving a nascent chain-ribosome complex. Hence, the mechanisms of targeting discrimination elucidated in this work prevent this co-translational targeting to the membrane. Inside host cells, TMD-substrates are not made by ribosomes but enter by injection through the T3SS, hence, there is no co-translational targeting mechanism to prevent. At this point it is entirely unclear whether host factors aid in targeting T3SS TMD-substrates to their target membrane. It is noteworthy, however, that most reported T3SS TMD-substrates (examples are all translocon components, Tir, SseF, SseG) insert into that membrane that is directly targeted by the T3SS injectisome, i.e. most TMD-substrates may not have to travel far to reach their destination.

As suggested, we have now extended our discussion of this point as follows (ll 426ff):

"Since TMD-effectors enter the host cell through injection, membrane targeting and insertion of these proteins are - unlike for ribosomally-made host proteins - post-translational processes. While evidence suggests that membrane insertion of translocon-type TMD-effectors is aided by the needle tip of the injectisome^{1,2}, it is not clear at this point whether host proteins aid in membrane targeting and insertion of non-translocon type TMD-effectors. It is striking that most investigated type III-secreted TMD-effectors

insert into the membrane that is directly targeted by the T3SS injectisome (e.g. all translocon components, Tir, SseF, SseG), however, the mechanism of membrane insertion may not involve lateral partitioning through the T3SS translocon as recent data for Tir suggest³ but cytoplasmic intermediates⁴. Hence, in addition to playing a role in targeting discrimination, the intermediate hydrophobic density of TMS of TMD-effector proteins may also serve to prevent aggregation of these proteins in the host cell cytoplasm while being sufficient to mediate association of their TMD with the interface of lipid bilayers of host cells and subsequent membrane insertion. Membrane insertion may also be facilitated by the mostly low complexity of the TMDs of secreted transmembrane proteins (Fig. 1b, Supplementary Fig. 5)."

We have also removed the first mentioning of this thought on membrane insertion in the host in l 242 (now l 270).

- Could this property be the reason why most pore-forming translocators have to rely on chaperoning by the T3SS needle tip for insertion into the host cell membrane?

The mechanisms of insertion of pore-forming translocators have not been studied extensively in vivo. By and large it seems that translocators can insert spontaneously into membranes in vitro but that the needle tip protein facilitates their insertion in vivo. Indeed, the latter might be necessitated by the comparably low hydrophobicity of translocators but it might also simply ensure the close proximity of needle tip and assembled translocon that is expected to be required for efficient translocation of substrates.

We have now mentioned that membrane insertion of translocon-type TMD-effectors is aided by the needle tip (ll 428ff) but we refrain from discussing this point in depth to not lose focus.

- Fig. S4 suggests that ΔG_{app} is not the only determinant that affects export (e.g. it looks like the SseF R67L/S78L mutant is more defective for export than the R67L/T72L mutant (S4D), even though the ΔG_{app} values (S4C) would predict otherwise. Clearly export is not the same as membrane insertion, but the authors should comment on this. In fact, the Fig. 4B data would argue that membrane insertion is not the reason for the drop in export. Could it have to do with the chaperone interaction?

We present evidence in this paper that hydrophobicity is one but not necessarily the only determinant that affects export. This is particularly clear for SseF, which gets targeted to the bacterial inner membrane in the absence of its chaperone SscB and whose inner membrane targeting is efficiently prevented also for TMS of stronger hydrophobicity. Indeed, SseF mutant S78L seems to exhibit a positional rather than just a physical effect on secretion. It is conceivable that S78L affects chaperone binding and consequently secretion. In fact, we do see an altered crosslinking profile in the S78L mutant (Fig. 5). We also see a reduced accumulation at the membrane that suggests a reduced association with the sorting platform of the injectisome.

We have now added the following statement in ll 264ff: "In agreement with this notion we observed that an increase in hydrophobicity of the first TMS of SseF resulted in a reduced secretion of these proteins into the culture supernatant, even though the reduction in secretion did not strictly follow the increase in hydrophobicity (Supplementary Fig. 4)." In addition, we addressed this point more extensively in the Discussion section (ll 480ff): "Our data on SseF show that several factors may contribute to targeting discrimination. SseF's cognate chaperone SscB prevented futile membrane targeting also when SseF contained TMS of strong hydrophobicity (Fig. 4B), however, the stronger hydrophobicity did still reduce secretion (Supplementary Fig. 4D). In particular for SseF mutant S78L it is conceivable that reduced secretion is a consequence of improper chaperone binding. In fact, we did observe an altered crosslinking profile between SseF_{S78L} and SscB (Fig. 5D). We also noticed a reduced accumulation of this mutant at the membrane (Fig. 4B), which may be the consequence of impaired targeting to the sorting platform of the injectisome, caused by improper chaperone binding."

Minor comments:

- Line 124, why list Escherichia coli, if all other bacteria are listed by genus name only?

This inconsistency has been corrected.

- Line 132, this seems a bit trivial since, at the very least, these T3SS will be secreting translocator proteins, which have TMs

We rephrased the sentence to: "Over all, this analysis illustrates that T3SS commonly secrete TMD-substrates with effector functions other than translocators and suggests..."

- In Fig. 3B the SseF64-85 blot seems cut off too close to the full length band, so that the truncated protein wouldn't be visible.

The figure has been corrected.

- Would it be possible to include the ΔG_{app} values for the various TM domains, e.g. in Fig. 2C, 3B/C, and importantly, Fig. 4B

The ΔG_{app} values have been included in the figures.

Reviewer #2 (Remarks to the Author):

The paper entitled "Revealing the mechanisms of membrane protein export by virulence associated bacterial secretion systems" focuses on the partitioning of hydrophobic membrane localization domains (MLDs) in proteins secreted via the type 3 and type 4

secretion systems. The authors show by bioinformatics analysis that MLDs in secreted proteins have lower hydrophobicity and "lower hydrophobic density" compared to the bona fide membrane proteins in *E. coli*. In addition, using a set of biochemical assays, they test the capacity of MLDs to integrate in the inner membrane and to bind to SRP. These experiments show that isolated segments of several T3SS substrates support partial membrane integration. Chaperon binding was assessed by the site directed UV crosslinking T3SS substrates. The authors elaborate a plausible model according to which intermediate hydrophobicity and chaperon binding near or at MLDs contribute to the targeting and T3S-mediated secretion of MLD-containing proteins.

Although their idea is not novel, the authors provide some new experimental support to their claims using several complementary approaches. I am not completely convinced by the quantitative aspects of their assessments.

Major comments:

1. The idea developed in this study is not novel, although it is presented as such. The authors fail to cite previous work and to acknowledge the large number of studies that discuss and support this idea. In 2006 the Cornelis group (Letzelter et al. *EMBO* 25, 3223–3233) showed that the *Yersinia* chaperon SycO binds to the YopO MLD in the cytosol and prevents its aggregation. Deletion of MLD renders YopO chaperon-independent for secretion and prevents its final membrane localization in a eukaryotic target cell, as shown by GFP fusions. Letzelter et al. even proposed that the chaperons have emerged originally to shield aggregation-prone MLDs, before acquiring a secondary role in targeting to the secretion apparatus.

We respectfully disagree with the reviewer in this point. The Cornelis group reported in the EMBO J paper by Letzelter et al.⁵ that SycO and other T3SS class-I chaperones bind membrane-localization domains of T3SS effectors that essentially coincide with the reported chaperone-binding domain of these T3SS substrates. They also report that this MLD-binding serves to prevent the intrabacterial aggregation of these substrates. The critical difference to the data and concept presented in our manuscript is that the MLD of the investigated substrates YopO, YopE, and YopT is not a transmembrane domain (TMD) but a domain that supports the peripheral membrane localization in host cells. This is substantially different to the membrane integration facilitated by TMDs. Because of the comparably low hydrophobicity of MLDs, the problem of intrabacterial targeting discrimination that is resolved by our work is not critical for these MLD-containing proteins.

Nonetheless, the reviewer is correct that our manuscript would benefit from a discussion of the concept introduced by Letzelter et al. and so we have included this into the revised version of our manuscript (ll 475ff): "Along the same lines, shielding of membrane localization domains that serve in targeting some T3SS effectors to the periphery of host membranes was also reported to involve engagement of cognate T3SS chaperones⁵. The comparably low hydrophobicity of these domains would not result in membrane targeting inside the bacterium but the cognate chaperones were shown to prevent intrabacterial aggregation of the respective effectors."

2. The membrane integration assay developed by von Heijne is used here to compare integration of several different peptides. It is very difficult to assess the results of this assay in a quantitative manner. The band of the full-length chimera never changes the intensity and there are no proper internal controls. In the pilot assay hydrophobicity of a single protein and its variants is compared, and even there the amount of the cleavage product is much higher than the amount of full-length membrane inserted protein. Minor differences in the "ratio" and the small amount of cleavage product appear to undergo important variations. Therefore, the summary of results in the form of a quantitative graph with error bars in my view is not very meaningful.

The reviewer raises a very valid point here that we apparently have not made sufficiently clear in the manuscript. Critical for the work done by Öjemalm and von Heijne was the ability to carefully and reliably quantify the fraction of inserted vs. translocated fragments of the Lep-LacY chimera. Hence, the authors of that paper performed all necessary controls and reported them in the paper. They reported a GlpG- and DegP-dependent processing of Lep-LacY into smaller fragments that necessitated a correction of the raw data for the degradation rates of all fragments of Lep-LacY (inserted, translocated, translocated_{small}). Using relevant constructs and mutants, they measured the degradation rates of all fragments by pulse chase analysis and developed an equation to correct for the different degradation rates. Concerning the never changed full-length band (Lep-LacY (T+I)), they stated: "Notably, Lep-LacY(Ts) is significantly more stable than the other forms, which may at least in part explain why there is little or no increase in the amount of Lep-LacY(T+I) corresponding to the decrease in LepLacY (Ts) as the hydrophobicity of the H segment is increased⁶.

The data graphed in Fig. 2C of the submitted manuscript were based on the uncorrected values directly obtained from Western blotting. The data shown in the revised version of the manuscript are now based on values corrected for the different degradation rates reported by Öjemalm and von Heijne. We have included a statement on the degradation in the materials and methods as well as in the figure legend of Fig. 2C.

In light of the critique of the reviewer, and as an exact quantitation is not critical for our assessment of insertion, we now omit the graph and the quantitation and only show numbers of approximate relative insertion below the Western blots.

We also realized that the cartoon in Fig. 2B showed the P2 domain of Lep at the wrong position. We have corrected this error in the revised version.

3. The proteolysis assays in Fig. 3B. show cropped images just below the full-length band and one could not see the proteolytic fragments for SipB 320-353 or SseF 64-85 if they were there. The reduction of full-length protein level is nevertheless an indication of cleavage. How do the authors interpret the behavior of Tir segment, which shows cleavage but at the same time similar protein levels? Results of the two assays do not seem to be entirely consistent. How do the authors explain the observed differences?

The too tight cropping of some images has been corrected.

The overall reduction in the levels of some chimeras upon proteinase K-treatment might result from post-translational targeting of these constructs to the bacterial periplasm, where they would be subject to digestion by proteinase K. It is noteworthy along these lines that the second TMS of Lep is not particularly hydrophobic ($\Delta G_{23} = 0.0$) and may be translocated if no interaction with its native interaction partner, namely the native first TMS of Lep, is possible.

We have now addressed this point in the text on page 8, ll 264ff.

The Lep-inv-Tir chimera shows two bands also in the proteinase K-untreated sample. Proteinase K does not change this appearance. Hence we believe that the two bands either result from unspecific degradation or that the chimera simply exhibits an abnormal running behavior upon SDS PAGE with partial unfolding, which is not uncommon for membrane proteins.

The charge distribution in these segments might influence membrane orientation, has this been taken into account in the predicted topology?

The reviewer raises an important point to consider when making these chimeras. The segments we selected did not contain charged residues and thus should not influence the topology of the test constructs. Only SseF₆₄₋₈₅ contains an Arg at position 67, which does, however, not seem to have a negative impact.

4. For several protocols and experiments the authors do not provide sufficient detail (e.g. urea extraction analysis). several others have been described in a very laconic manner.

According to the guidelines of Nature Communications, we have now extended the Methods section and avoided the use of reference to previous publications.

Minor comments.

1. It would be helpful if the lines in the text were numbered continuously.

The submitted version that we have at hand does show a continuous numbering.

2. The authors should avoid laboratory jargon. The legend of Figure 2 mentions the Keio ref strain - the strain has its name, BW25113. Also, what is a skipped TMS?

The strain BW25113 has been termed correctly in the revised manuscript.

The term "skipped TMS" was coined by Bukau and Kramer in Schibich et al. (2016)⁷, to which we refer extensively. We have attempted to better clarify the term in the revised manuscript and write now "... N-terminal TMS skipped by SRP of otherwise SRP-targeted proteins... in line 246 of the manuscript.

Reviewer #3 (Remarks to the Author):

Major Comments

Krampen et al report on the secretion of T3SS and T4SS membrane proteins in pathogenic bacteria and shed light on how these proteins avoid being mistargeted to the Sec translocon and inserted into the membrane. The overall results show that they do so typically by possessing a weakly hydrophobic TM segment that can integrate into the membrane but is insufficiently hydrophobic to allow targeting to SRP which would direct the protein to the Sec translocon for membrane insertion. In addition, they find that many of the T3SS membrane proteins contain cognate chaperones (that bind to the amino-terminal cytoplasmic domain of the substrate and the TM segment) that prevent targeting by SRP. They propose a two-pronged mechanism, namely low hydrophobicity of the TM segments and targeting by cognate chaperones, that explains how substrates can be accurately routed for type III secretion. While I have some concerns with some of the studies and interpretations, I think the results are quite interesting and advance the field.

The authors used several different approaches to study the type 3 secretion membrane proteins. First, the authors use bioinformatics to show that out of 174 T3SS known substrates there are 37 membrane protein candidates, typically with one or two TM segments (Fig. 2A). The transmembrane segments of the T3SS generally have lower overall hydrophobicity than single *E. coli* TM membrane proteins. Several candidate TM segments (derived from the type 3 secretion substrates SipB, SseF and Tir) were tested in *E. coli* using a chimeric construct where a test segment (placed in between two known TM segments) can be examined for membrane integration using a previous established assay involving GlpG cleavage (Fig. 2B). The authors discovered each of the TM segments of the type 3 secretion substrates were membrane integrated. Second, they showed only one of these TM segments could promote insertion of inverted Lep when the T3SS TM segment was interchanged with the H1 domain of inverted Lep (Fig. 3B). Similarly, the tested TM segments of T3SS could not promote translocation of the amino-terminal domain of a ProW-P2 construct when substituted for the proW first TM segment (Fig. 3C). They suggest this is because the TM segment was insufficiently hydrophobic to promote SRP targeting.

The authors next examine the authentic type 3 secretion substrate SscF (Fig. 4 and Fig. S3) and SipB (Fig. S3) in *Salmonella* and show that the substrates are secreted into the supernatant but only when the cognate chaperone SscB is present in the cell (Fig. 4). When the cognate chaperone was not present, the membrane substrate SipB was not integrated into the membrane, while SseF was membrane integrated (Fig. S9). However, SseF was not membrane integrated (Fig. S3 D and E) when the chaperone SscB was present; presumably because it prevents SRP targeting to the Sec translocon. Finally, they show, using photocrosslinking studies, that the chaperone SscB interacts with SseF substrate since it can be crosslinked to SseF when a photoprobe is placed in either the chaperone binding domain or in the first TM domain.

In addition, the authors also examined Type 4 secretion substrates. These substrates, which are secreted, can also be a membrane protein. Similar to Type 3 secretion substrates, they observed the TM segments of type 4 secretion membrane substrates had a lower hydrophobicity than single TM E. coli proteins. Of the four tested TM segments of T4SS proteins, only one could function in place of the first TM segment of inverted Lep and promote membrane insertion. The authors suggest that generally the TM segment of T4SS membrane proteins cannot promote SRP targeting (although one of them could).

In conclusion, the work reveals how secretion systems can specifically secrete membrane protein substrates and avoid integrating them into the plasma membrane, prior to secretion out of the cell.

We thank the reviewer for this extensive appreciation of our work.

Minor Comments

1. On the membrane targeting potential of transmembrane segments of T3SS protein, the authors suggest that the amino-terminal TM segments of the type 3 secretion substrates can promote insertion of the inverted Lep only when it can initiate SRP targeting. The idea is that the TM segments fail to promote membrane insertion due to the fact that they cannot be targeted to SRP. It might be best to show the TM segments that fail to promote insertion do not bind to SRP but the one candidate that does (SecF 86-104), indeed, binds to SRP. Photocrosslinking could be used to answer this question.

We agree with the reviewer that a formal prove of SRP binding may be a good addition to this story. Unfortunately, while in vitro photocrosslinking with stalled ribosome-nascent chains was the method of choice to experimentally address SRP-binding in earlier days, it is not any more regarded as reliable to address these targeting events in the light of their highly kinetic nature. SRP was shown to bind incorrect targets and to release them again at later checkpoints, so SRP-binding does not equal SRP-targeting⁸. Also in vivo photocrosslinking as used by us to study the interaction of SseF and SscB is not well suited to study transient interactions like the one between SRP and its substrates. For these experimental limitations, we would like to refrain to provide experimental prove for the differential SRP binding between the substrates of different hydrophobicity but solely refer to the bioinformatics assessment based on the studies by Schibich et al.⁷ and Hessa et al.⁹. We believe that the studies by Schibich et al. and Hessa et al. allow for a very solid judgement of the membrane targeting and insertion potential of hydrophobic protein segments and so the experimental assessment of SRP binding of model proteins may not add much information, in particular if the methodology is not well suited. Rather, the evaluation of co-translational targeting events of secreted membrane proteins inside Salmonella or Legionella should be pursued by extensive follow-up studies. Also, we are aware that SRP is not the only membrane targeting pathway in the bacterial cell. We toned down the mentioning of SRP at several positions in the manuscript (I113, I270) since we have not formally proven the involvement of SRP for the proteins tested.

2. The authors should address why they observed a significant decrease in the SipB 320-353 construct when protease is added in Fig. 3B. Is this due to lysis even though band X does not decrease? Similar concern for the proW study. There is a decrease in SipB 320-353 as well as SipB 320-337 when proteinase K is added (Fig. 3C).

The missing lower band in response to proteinase K digestion shows that SipB 320-353 is not targeted to and integrated into the membrane, which is what we wanted to assess. The overall reduction of this as well as the ProW-based constructs upon proteinase K digestion may result from post-translational targeting of these constructs to the bacterial periplasm, where they would be subject to digestion by proteinase K. It is noteworthy along these lines that the second TMS of Lep is not particularly hydrophobic ($\Delta G_{23} = 0.0$) and may be translocated if no interaction with its native interaction partner, namely the native first TMS of Lep, is possible.

We have now addressed this point in the text on page 8, ll 264ff.

3. On page 4, it is stated (sentences 292-294) that an increase in hydrophobicity of the first predicted TMS of SseF well beyond the SRP-targeting threshold did not result in erroneous membrane targeting in the presence of SscB (Fig. 4B). On page 8, it is stated “that a lower hydrophobicity of the TM segment prevents futile SRP-pathway targeting. In support of this, an increase in hydrophobicity of the first TMS of SipB and SseF, respectively, resulted in a reduced secretion of these proteins into the culture (Fig. S4)”. At first glance this seems contradictory. Please clarify what you mean. Moreover, I would recommend showing directly that when the TM segment is made more hydrophobic that the protein can be integrated into the membrane.

The prevalence of the intermediate hydrophobicity of TMS of T3SS TMD-substrates is striking and based on our knowledge of SRP-targeting⁷ and membrane insertion⁹, our hypothesis that the intermediate hydrophobicity aids in substrate discrimination between membrane targeting and type III secretion is well conceivable. However, we present evidence in this paper that hydrophobicity is one but not necessarily the only determinant that affects export. This is particularly clear for SseF, which gets targeted to the bacterial inner membrane in the absence of its chaperone SscB and whose inner membrane targeting is efficiently prevented also for TMS of stronger hydrophobicity.

We do show that an increase in hydrophobicity does not lead to increased membrane association of SseF in the presence of its chaperone SscB. Since SseF already inserts into the membrane in the absence of SscB, we omitted to show the membrane integration for SseF mutants of stronger hydrophobicity. Because of this behavior, SseF cannot serve to directly show “that when the TM segment is made more hydrophobic that the protein can be integrated into the membrane”.

Unfortunately, SipB did not serve to pin down this point either since mistargeted material seems to be very quickly degraded. It has been in fact known for long that SipB is highly unstable in the absence of its chaperone SicA¹⁰.

We have now extended our discussion of this point in ll 480ff to improve the clarity.

4. In the Discussion, line 337, I would tone this down since one of the candidate TM segments of type 4 secretion membrane protein did promote membrane targeting (LegC3) (Fig. S3). Similarly, tone down line 398, where it is stated “that type IVB-secreted transmembrane proteins possess TMS of low hydrophobic density and it is comprehensible that this passive mechanism suffices for efficient targeting discrimination”. Based on the author’s data, it is not clear why the T4SS containing this LegC3 TM segment avoids membrane integration. This should be discussed.

Indeed, one out of four tested T4BSS TMD-substrates was shown to be targeted to and inserted into the inner membrane. However, based on the 75% whose hydrophobicity did not suffice to promote membrane targeting, we feel still comfortable with our argument. Also the distribution of the ΔG_{18-35} and ΔG_{12-17} minima in T4BSS TMD-substrates tells that the majority of these substrates is unlikely to be targeted to the inner membrane. Nonetheless, we did tone down the statement now in line 394 as suggested by the reviewer to take into account that a few T4BSS substrates are known to exist as inner membrane intermediates before being secreted into the host cell. We discussed this in lines 494ff.

Lines 394ff now reads: “This mechanism of passive targeting-avoidance may be a general principle, which we also identified for a subset of substrates of T4BSS of Legionella and Coxiella.”

We also toned down the statement now in line 491: “We show here that many type IVB-secreted transmembrane proteins possess ...”

Page 12 (line 355), I would change “In consequence” to “Consequently”.10

The sentence has been changed as suggested.

References:

1. Marenne, M.-N., Journet, L., Mota, L. J. & Cornelis, G. R. Genetic analysis of the formation of the Ysc-Yop translocation pore in macrophages by *Yersinia enterocolitica*: role of LcrV, YscF and YopN. *Microb. Pathog.* **35**, 243–258 (2003).
2. Veenendaal, A. K. J. *et al.* The type III secretion system needle tip complex mediates host cell sensing and translocon insertion. *Mol Microbiol* **63**, 1719–1730 (2007).
3. Mao, C. *et al.* Translocation of enterohemorrhagic *Escherichia coli* effector Tir to the plasma membrane via host Golgi apparatus. *Mol Med Rep* **16**, 1544–1550 (2017).
4. Race, P. R., Lakey, J. H. & Banfield, M. J. Insertion of the enteropathogenic *Escherichia coli* Tir virulence protein into membranes in vitro. *J Biol Chem* **281**, 7842–7849 (2006).
5. Letzelter, M. *et al.* The discovery of SycO highlights a new function for type III secretion effector chaperones. *EMBO J* **25**, 3223–3233 (2006).
6. Ojemalm, K., Botelho, S. C., Stüdle, C. & Heijne, Von, G. Quantitative analysis of SecYEG-mediated insertion of transmembrane α -helices into the bacterial inner membrane. *J Mol Biol* **425**, 2813–2822 (2013).
7. Schibich, D. *et al.* Global profiling of SRP interaction with nascent polypeptides. *Nature* **536**, 219–223 (2016).
8. Zhang, X., Rashid, R., Wang, K. & Shan, S.-O. Sequential checkpoints govern substrate selection during cotranslational protein targeting. *Science* **328**, 757–760 (2010).
9. Hessa, T. *et al.* Molecular code for transmembrane-helix recognition by the SecE1 translocon. *Nature*

- 450**, 1026–1030 (2007).
10. Tucker, S. C. & Galan, J. E. Complex function for SicA, a *Salmonella enterica* serovar typhimurium type III secretion-associated chaperone. *J Bacteriol* **182**, 2262–2268 (2000).

REVIEWERS' COMMENTS:

Reviewer #2 (Remarks to the Author):

The authors have addressed all my comments to the best of their abilities and I am in favor of publication of the manuscript in its revised form.

Reviewer #3 (Remarks to the Author):

The authors have satisfactorily addressed my concerns in the first round of the review process. The revised paper makes an important advance in understanding how type III and type IV membrane protein effector proteins avoid targeting to the bacterial membrane.